# miR-1183 Is a Key Marker of Remodeling upon Stretch and Tachycardia in Human Myocardium

**DOI:** 10.3390/ijms23136962

**Published:** 2022-06-23

**Authors:** Natasa Djalinac, Ewald Kolesnik, Heinrich Maechler, Susanne Scheruebel-Posch, Brigitte Pelzmann, Peter P. Rainer, Ines Foessl, Markus Wallner, Daniel Scherr, Akos Heinemann, Simon Sedej, Senka Ljubojevic-Holzer, Dirk von Lewinski, Egbert Bisping

**Affiliations:** 1Department of Internal Medicine, Division of Cardiology, Medical University of Graz, 8036 Graz, Austria; natasa.djalinac@medunigraz.at (N.D.); ewald.kolesnik@medunigraz.at (E.K.); peter.rainer@medunigraz.at (P.P.R.); daniel.scherr@medunigraz.at (D.S.); simon.sedej@medunigraz.at (S.S.); senka.ljubojevic@medunigraz.at (S.L.-H.); egbert.bisping@medunigraz.at (E.B.); 2Unit of Human Molecular Genetics and Functional Genomics, Department of Biology, University of Padua, 35121 Padua, Italy; 3Department of Cardiothoracic Surgery, Medical University of Graz, 8036 Graz, Austria; heinrich.maechler@medunigraz.at; 4Gottfried Schatz Research Center, Institute of Biophysics, Medical University of Graz, 8010 Graz, Austria; susanne.scheruebel@medunigraz.at (S.S.-P.); brigitte.pelzmann@medunigraz.at (B.P.); 5BioTechMed Graz, 8036 Graz, Austria; 6Department of Internal Medicine, Division of Endocrinology and Diabetology, Medical University of Graz, 8010 Graz, Austria; ines.foessl@medunigraz.at; 7Cardiovascular Research Center, Lewis Katz School of Medicine, Temple University, Philadelphia, PA 19140, USA; 8Institute of Experimental and Clinical Pharmacology, Medical University of Graz, 8010 Graz, Austria; akos.heinemann@medunigraz.at; 9Institute of Physiology, Faculty of Medicine, University of Maribor, 2000 Maribor, Slovenia

**Keywords:** stretch, tachycardia, cardiac remodeling, microRNA, *MIR1183*, miR-1183, human myocardium, biomarker

## Abstract

Many cardiac insults causing atrial remodeling are linked to either stretch or tachycardia, but a comparative characterization of their effects on early remodeling events in human myocardium is lacking. Here, we applied isometric stretch or sustained tachycardia at 2.5 Hz in human atrial trabeculae for 6 h followed by microarray gene expression profiling. Among largely independent expression patterns, we found a small common fraction with the microRNA miR-1183 as the highest up-regulated transcript (up to 4-fold). Both, acute stretch and tachycardia induced down-regulation of the predicted miR-1183 target genes *ADAM20* and *PLA2G7*. Furthermore, miR-1183 was also significantly up-regulated in chronically remodeled atrial samples from patients with persistent atrial fibrillation (3-fold up-regulation versus sinus rhythm samples), and in ventricular myocardium from dilative cardiomyopathy hearts (2-fold up-regulation) as compared to non-failing controls. In sum, although stretch and tachycardia show distinct transcriptomic signatures in human atrial myocardium, both cardiac insults consistently regulate the expression of miR-1183 and its downstream targets in acute and chronic remodeling. Thus, elevated expression of miR-1183 might serve as a tissue biomarker for atrial remodeling and might be of potential functional significance in cardiac disease.

## 1. Introduction

Atrial remodeling is the response of the atria to a variety of acute insults or chronic load. Its consequences have major clinical impact associated with decline in hemodynamic and valvular function, the generation of thromboembolism and stroke, and the development of atrial fibrillation with significant individual morbidity and also enormous socioeconomic burden [1].

The remodeling process starts with excitation transcription coupling and altered gene expression as first steps followed by functional and structural changes, including atrial enlargement, altered geometry and sphericity, and altered extracellular matrix and atrial fibrosis [2]. In human heart disease this is, typically, a result of several combined triggering mechanisms which can potentiate each other and result in remodeling progression, e.g., atrial fibrosis will increase the probability of atrial fibrillation, and atrial fibrillation triggers progression of atrial fibrosis [3].

Though different in origin, most of the triggering mechanisms can be categorized to one of these two categories: 1. Stretch, and 2. Tachycardia. Stretch is the cellular correlate to volume overload, and, if combined with pure isometric contraction, it also corresponds to pressure overload. Those load conditions may be found as a result of valvular stenosis or regurgitation, diastolic dysfunction with or without cardiac hypertrophy, heart failure with either preserved (HFpEF) or reduced ejection fraction (HfrEF), arterial or pulmonary hypertension, and others [4]. In contrast, tachycardia corresponds to either paroxysmal episodes of atrial runs, atrial tachycardias, atrial flutter, or atrial fibrillation or persistent to permanent forms of tachycardias [5].

In chronic human atrial remodeling, the contribution of each of these mechanisms is unknown, which makes it difficult to therapeutically target the specific mechanisms. Data on trigger-specific remodeling and their acute initiating phase in human myocardium is lacking. Thus, we performed experiments under controlled conditions in isolated human atrial myocardium. We show that stretch and tachycardia induce very different early remodeling patterns but share up-regulation of the microRNA, miR-1183. We also present putative downstream targets of miR-1183 which are involved in the response to both triggers. And we show that this microRNA is not only regulated in an acute setting but also in chronic remodeling, and we discuss its potential causative role in cardiac remodeling.

## 2. Results

### 2.1. Stretch and Tachycardia Induced Differential and Common Patterns of Gene Expression

Muscle strips (trabeculae) isolated from the right atrial appendage of heart surgery patients (Table 1) were subjected to two triggers of early remodeling: sustained stretch with increased pre- and afterload for 6 h or sustained tachycardia stimulation (2.5 Hz) for 6 h. Microarray screening was performed with five patient samples for each of these four groups: 1. Stretch, 2. Control for stretch, 3. Tachycardia, and 4. Control for tachycardia. Original data files are uploaded to the GEO repository under the series record GSE135166.

The microarray transcripts could be classified in four groups with a similar distribution for the stretch- and the tachycardia-regulated genes: protein-coding (approx. 60% in both), predicted protein-coding (2–3%), non-coding including small RNAs (18–22%), and transcripts without a functional annotation (approx. 20%). The analysis for protein coding and non-coding genes with *p* < 0.05 showed a largely independent regulation by stretch versus tachycardia as demonstrated by 1305 transcripts regulated solely after stretch and 1837 solely after tachycardia (Figure 1A), which also revealed that tachycardia resulted in a 40% higher number of significantly regulated genes than stretch. Out of the 1305 stretch regulated genes, 683 were found to be up-regulated and 622 down-regulated. Similarly, tachycardia resulted in an equal ratio of 922 up-regulated and 915 down-regulated transcripts. The list containing all significantly regulated transcripts for either trigger conditions sorted by gene symbol can be found in Appendix A. Among the protein coding genes *ACTA1* was the highest up-regulated gene after 6h of pathological stretch (Figure 1C), while *TRAJ22* and *SNRPN* were the highest up-regulated genes after sustained tachycardia (Figure 1B). qPCR assessment of a selection of best up-/down-regulated genes and genes with a known cardiovascular role (*CNN1*, *PLA2G7*) on the microarrays confirmed a similar extend of array and qPCR expression in the same samples (Figure 1D).

Despite the predominantly independent gene regulation between stretch and tachycardia, there were 65 transcripts that were commonly regulated by both triggers (Figure 2A): The larger fraction (39 genes) displayed unidirectional regulation (Figure 2B) while the smaller fraction (26 genes) was regulated in opposite direction (Figure 2C) between both triggers. The majority of the common genes with identical direction were up-regulated in their expression. Among characterized transcripts with a known biological function a microRNA encoding transcript, *MIR1183*, had the highest expression level. In contrast, *MIR4501, ADAM20,* and *PLA2G7* were the most down-regulated common transcripts. A complete list of all significantly regulated transcripts and all 65 commonly regulated transcripts is provided in the Appendix A.

Bioinformatic estimations via ingenuity pathway analysis (IPA) provided information on molecules and associated networks, canonical pathways, disease and functions, upstream regulators, and causal networks which are summarized in Appendix A. By these means we also determined potential upstream regulators of the affected target molecules on our arrays: for stretch this included inhibition of PARP1 and BRD4 and activation of MAP2K1/2. In contrast, tachycardia pointed to a larger spectrum of potential upstream regulators with several activated factors (TP53, RBM5, JAG2, PDGF BB, NR3C2, SPP1, and IL1RN) and even more inhibited factors (TNF, JUN, IKBKB, IL4, TLR4, IL1A, CD40, IL18, PRKCD, P38 MAPK, Lymphotoxin, PTGS2, CD14, C5, Gm-csf, MBD2, Fcer1, IL17A, Jnk, ERK1/2, SAA, TP63, ECSIT, NDRG1, ESR1, and Mek).

### 2.2. MicroRNA Expression Profile and Detection of miR-1183 Encoding Gene as the Strongest and Commonly Regulated Transcript

After evaluation of all protein coding genes, we generated a list of all significantly regulated miRNA genes (Figure 3A). Small RNAs, including miRNAs, were present in a close to equal relative expression of 13% and 14% from total regulated stretch and tachycardia regulated transcripts, respectively. Interestingly, stretch showed an almost exclusive up-regulation of affected miRNAs. Conversely, tachycardia stimulation induced a balanced response concerning miRNA expression (Figure 3A). Among the hits, *MIR1183* was the top regulated transcript not only among miRNAs but also when considering overall array gene expression. More importantly, *MIR1183* was the highest regulated transcript that involved both stretch and tachycardia (Figure 3A). Accordingly, qPCR evaluation confirmed the up-regulation to be significant with stretch and a strong trend with tachycardia (Figure 3B).

### 2.3. miR-1183 as a Biomarker in Human Cardiac Tissue

Considering the novelty of microarray findings, we further investigated whether miR-1183 is also regulated in chronic remodeling and might serve as a tissue biomarker. Therefore, we evaluated chronic gene expression levels in samples of both atrial and ventricular source. Samples from patients suffering from documented long-term atrial fibrillation were compared to those with stable sinus rhythm (Table 2).

Additionally, ventricular samples from patients suffering from end-stage dilated cardiomyopathy (DCM) were compared to samples from non-failing donor hearts (NF) (Table 3).

Atrial myocardial samples from patients with chronic atrial fibrillation showed a significant increase in pri-miR-1183 levels compared to patients with normal sinus rhythm (Figure 4A). Samples from patients with dilated cardiomyopathy (DCM) also showed a significant up-regulation compared to samples from non-failing donor hearts (Figure 4B).

### 2.4. Bioinformatics Prediction of miR-1183 Downstream Targets

For investigation of miR-1183 mechanism of action we used the tool TargetScan7.2 to assess miR-1183 target genes. Out of the complete list of possible target genes (3539) with a different cumulative weighted context ++ score which predicts efficacy of miRNA binding to complementary mRNA sites [6] 1337 were expressed on the microarray and are summarized in Appendix A. We looked solely at down-regulated genes, since down-regulation reflects direct miRNA effects on mRNA. Out of these target genes, 24 down-regulated genes emerged under tachycardia (Figure 5A) whereas 13 showed significant down-regulation under stretch (Figure 5B). The most interesting candidate mRNA targets include the stretch and tachycardia commonly expressed genes *ADAM20* and *PLA2G7* (Figure 5C). Additionally, both *ADAM20* and *PLA2G7* are the protein coding genes with the highest common down-regulation detected on the microarray and confirmed via qPCR.

## 3. Discussion

Our results show that the pathological triggers, stretch and tachycardia, induced largely independent early remodeling responses in human atrial myocardium. Both triggers shared only a small common fraction, but, within this fraction, the pri-microRNA encoding transcript, *MIR1183**,* was the strongest up-regulated for both on genome-wide level. Furthermore, we demonstrated that miR-1183 is not only up-regulated acutely but also in chronic cardiac remodeling in both human atrial and ventricular tissue. As downstream targets of miR-1183 we identified *ADAM20* and *PLA2G7*. Our data characterized miR-1183 as novel tissue biomarker in human cardiac remodeling with a potential functional cardiac role.

### 3.1. Microarray Gene Expression Uncovers Independent Stretch and Tachycardia Gene Regulation

With our experimental settings we were able to uncover trigger-specific remodeling mechanisms which are otherwise masked by each other in vivo. This is the first study to allow those insights in human myocardium. We found that stretch and tachycardia induced very independent genome-wide expression profiles, and that tachycardia had a 40% higher impact on the number of transcripts at the very early time point. A recent study focused on the chronic response in human atrial appendage samples from patients with right atrial dilatation but no history of atrial fibrillation suggesting stretch without tachycardia as a remodeling trigger in these hearts [7]. The authors found a rather small number of stretch regulated genes by genome-wide profiling, which fits to our findings in comparison to more regulated genes with tachycardia. Also, there was an overlap between their and our microarray datasets for several stretch regulated genes (*DEFB109P1B, LOC102723337, LOC101927029, RHPN2, CD177,* and *LYPD6*), but also for tachycardia regulated genes.

Some of the highest stretch up-regulated genes in our dataset are well known in cardiac disease: skeletal alpha (α)-actin, *ACTA1*, is involved in regulation of muscle contraction [8] as well as cytoskeleton organization [9]. *ACTA1* expression in the cardiovascular systems can serve as a marker of pathological hypertrophy [10,11,12,13,14], associated cardiac diseases [15,16], and, ultimately, heart failure. The *CNN1* gene encodes Calponin 1, which expression was previously shown to have a stretch-mediated up-regulation pattern [17,18]. The gene *TP1* encodes an enzyme participating in processes of glycolysis and gluconeogenesis and has a role in skeletal muscle myopathy [19] and metabolic regulation in type 2 diabetes hearts [20].

We also found tachycardia up-regulated genes with cardiac relevance: *SNRPN* is a human gene that encodes the small nuclear ribonucleoprotein polypeptide N with a role in development [21,22] and detected single nucleotide polymorphism in COPD patients and participants of the Framingham study [23]. *RCAN1* is linked to the calcineurin/NFAT signaling pathway [24,25,26]. It inhibits Calcineurin by binding to its catalytic domain and therefore represses pathological hypertrophy [27]. High expression of *RCAN1* is part of a coping mechanism against sudden insult in the brain [28] and lung [29] and is highly involved in regulating pathological cardiac remodeling either directly as mentioned previously or through interaction with other signaling mediators [30].

Our analysis by IPA was not able to predict any highly relevant signaling pathways tied to either stretch or tachycardia. However, it became obvious that tachycardia resulted in down-regulation of a number of genes involved in the process of inflammation, such as *IL1A**, IL1B, VCAM1, KIR3DL3,* and *CCL4*. As determined by the GO molecular function, interleukin 1 alpha and beta belong to a major pro-inflammatory cytokine family. VCAM1 is an adhesion molecule whose expression is additionally modulated by *IL1B* [31], *KIR3DL3* is involved in antigen processing, and CCL4 is a chemokine for a number of inflammatory cells.

Following the observation that stretch and tachycardia show these distinctive patterns of gene regulation, we next concentrated on the common gene fraction with unidirectional regulation and identified a non-coding miR-1183 precursor gene—*MIR1183,* which stands out as the top up-regulated array transcript with a predefined function. Consequently, we analyzed the expression changes of *MIR1183* and all other microRNA coding transcripts which were represented on our microarrays.

### 3.2. MicroRNA Regulation in Stretch Versus Tachycardia Induced Remodeling

Stretch-mediated miRNAs up-regulation was more prominent than down-regulation on our microarrays, and several of the up-regulated miRNAs had an established role in cardiac remodeling:

miR-378 acts as an antihypertrophic factor, is specific to cardiomyocytes, and targets components of the MAPK pathway in vitro. In vivo it prevents pressure overload induced hypertrophy in mice and improves cardiac function [32].

miR-216A is up-regulated in CAD patients and is involved in inflammation via the Smad3/IƙBα axis [33]. The expression of miR-216 also promotes cardiac fibroblast proliferation and biogenesis through the SMAD signaling pathway and autophagy inhibition [34]. Finally, it is up-regulated in plasma samples in heart-failure patients [35,36].

miIR-641 is involved in vascular smooth muscle cell proliferation and metastasis [37] and miR-16-1 in cytostatic cardiac injury [38].

The detailed cardiac function of several array expressed miRNAs is yet to be determined. However, in previous reports their up-regulation has been associated with different types of cardiac disease: miR-376C with females with gestational hypertension [39], miR-4268 with aortic stenosis [40], miR-330 with post-MI heart failure [41], and miR-4487 with CVD patients with “blood stasis syndrome” [42].

A few tachycardia-regulated miRNAs could also be assigned to a cardiovascular role: miR-221 is increased after intense sport activity and associated with PW and CK biomarkers [43] which is consistent with increased workload after tachycardia. miR-519b is found to be up-regulated in AFIB patients [44] while miR-518e in non-ischemic heart-failure [45].

In contrast to stretch, tachycardia-mediated down-regulation appears more prominent and in positive correlation with cardiac remodeling: down-regulation of miR-599 is in relation with cardiomyocyte sensitivity to oxidative stress and, eventually, pyroptosis [46] and is also consistent with the direction of regulation in our array. Apoptotic cell death is linked to low expression of miR-498 in MI patients [47] and is in agreement with the down-regulation in our array. MiRNAs with a negative correlation between cardiac remodeling and the direction of regulation in our array, such as miR-28 [48,49], miR-374c [50], and miR-125a [51] are likely part of an early coping mechanism.

The common fraction of stretch and tachycardia regulated genes contained MIR1183 as the strongest up-regulated and MIR4501 as the strongest down-regulated microRNA transcripts. Whereas miR-4501 has no assigned cardiac function, the relevance of increased miR-1183 expression levels in cardiovascular disease has been previously established in a cohort of patients with rheumatic heart disease [52]. The authors observed high levels of plasma miR-1183 and a positive trend regarding tissue expression in the papillary muscle. Another study found that miR-1183 is among the top up-regulated microRNAs in plasma of patients diagnosed with essential hypertension [53]. However, direct assessment of miR-1183 levels in the human myocardium have not been reported yet, nor was the behavior upon an acute insult such as in our settings. MiR-1183 has no known orthologues in lower species, which might provide explanation for the lack of previous studies. The microarray gene chip can detect a gene encoding the precursor of the mature form of miR-1183 and, so, we aimed to verify this finding via qPCR and found a strong correlation between the array and qPCR expression levels.

### 3.3. miR-1183 as a Novel Marker in Atrial and Ventricular Remodeling

The atrial tissue samples from patients undergoing mitral valve replacement surgery showed atrial dilatation and persistent atrial fibrillation compared to the control group with only moderate dilatation and permanent sinus rhythm. Thus, this reflects increased atrial wall stress (i.e., cellular stretch) and tachycardic atrial rate as a remodeling trigger in the atrial fibrillation samples. We observed a significantly increased miR-1183 in these samples, which coincides with a previous study that found some other differentially expressed miRNAs in left and right tissue samples of atrial fibrillation patients [54].

We were also interested in seeing if the chronic miR-1183 expression changes in atrial myocardium were reflected in ventricular myocardium as well and found significant up-regulation of miR-1183 in dilated cardiomyopathy (DCM) compared to non-failing hearts. This finding confirms the value of miR-1183 as a tissue biomarker from the early onset of acute myocardial stress, towards chronic disease in the atrium and, finally, end-stage heart failure in the ventricle.

### 3.4. Upstream Regulators of miR-1183

IPA bioinformatics screening did not offer hints about possible miR-1183 upstream regulators. Thus, it remains open by what signaling cascade miR-1183 might be part of in human myocardium.

### 3.5. Downstream Targets of miR-1183 and Cardiac Functional Effects

We also performed a TargetScan search for all potential miR-1183 downstream targets and looked up those with suppressed gene expression on the array. Thus, an association has been established between several stretch-specific miR-1183 targets and cardiac remodeling, such as: KCNA3 with diabetic cardiomyopathy [55], MED23 with CHD [56], SAMSN1 with atherosclerosis [57], CCND1 with myocardial ischemia-reperfusion injury [58,59]*,* and MCOLN3 with atrial fibrillation [60].

Also the following tachycardia-specific miR-1183-regulated targets showed a cardiovascular association: the gene NEU3 with cardiac fibrosis and CHD [61,62], ZNRD1 with CHD [63], TRIM16 [64] with cardiac hypertrophy, BIRC3 with ischemic preconditioning [65], ARRDC3 with cardiac hypertrophy [66]*,* and MTAP with ischemic stroke [67].

Furthermore, we found two downstream targets in the common fraction of stretch and tachycardia regulated genes: *ADAM20* and *PLA2G7*:

*ADAM20* is a member of the dysintegrin and matrix metalloprotease domain family which is involved in processes of adhesion and proteolysis and is important for cell-to-cell and cell-to-matrix communication [68]. So far, *ADAM20* was found to have a role in managing cell fusion during reproduction [69] but a possible relationship with cardiovascular disease is not defined.

*PLA2G7* encodes a component of the protein LP-phospholipase 2 (LPA2) which mediates cardiac effects through an inflammatory response [70]. It is commonly down-regulated in foam cells in arteriosclerosis [71]. Also, down-regulation of *PLA2G7* has been shown to delay cardiac aging after caloric restriction [72]. LPA2 inhibition has been attributed to anti-inflammatory and anti-fibrotic benefits seen in a model of hypertensive cardiac disease [73]. By down-regulating *PLA2G7* and, consequently, reducing LPA2 expression miR-1183 might display a protective role against inflammatory cardiac remodeling, cardiac aging, and fibrosis. Since 1. additional inflammation mediators are down-regulated on our microarrays, and 2. IPA indicates involvement of the LXR/RXR nuclear pathway, we can speculate that cardiomyocytes might display a coping mechanism mediated through this pathway.

In addition to our microarray findings, it was recently published that Bcl2 is a direct molecular target of miR-1183 in rheumatic heart disease [74]. In our dataset *BCL2* and associated anti-apoptotic pathways were not significantly regulated, but its interaction partner *BNIP3* was down-regulated. *BNIP3* is a pro-apoptotic gene and correlated with progression towards heart failure [75,76], so that its down-regulation might point to a potential early coping mechanism upon stretch.

## 4. Material and Methods

### 4.1. Preparation, Processing, and Stretch and Tachycardia Induction in Human Myocardium

Human atrial muscle strips were isolated from right atrial appendages obtained from patients undergoing cardiac surgery after written consent was obtained. Ventricular muscle strips and tissue samples were obtained as biopsies from non-failing or failing hearts from multiorgan donors denied for transplantation due to medical reasons, such as advanced age, systolic or diastolic dysfunction, valvular disease, or multivessel coronary disease.

The isolated muscle strips (trabeculae) were mounted on hooks and kept in an oxygenized (95% O_2_; 5% CO_2_) organ bath setup (Scientific Instruments, Gilching, Germany) at 37 °C in Tyrode solution similar to what has been described previously [77]. For stretch induction, muscle strips were stimulated at 1 Hz frequency and stretched to the length of maximal isometric force (Lmax) while the control muscle strips were kept without preload on slack length. For tachycardia induction, the stimulation was set to 2.5 Hz frequency in duration of 6h and compared to a control at 1 Hz frequency. In addition, both groups were kept on slack length. Myocardial tissue samples were homogenized and used for RNA isolation. The patient’s medical history was assessed to determine a possible correlation between confounding factors with experimental outcome and gene expression dataset (Table 1 and Table 2).

### 4.2. RNA Isolation

Total RNA purification of all human specimens was performed using the miRNeasy on column isolation kit (Qiagen, Hilden, Germany) with DNase treatment. RNA yield was measured using the NanoDrop 2000 (Thermo Fisher Scientific, Wilmington, NC, USA). RNA quality was validated by the 2100 Bioanalyzer using the Eukaryote total RNA 6000 Pico assay (Agilent, Santa Clara, CA, USA) and the QiaXpert (Qiagen, Hilden, Germany) setup The samples were further processed for microarray hybridization or used for qPCR analysis.

### 4.3. Microarray Hybridization

Microarray screening was performed on a total of 20 arrays with five patient samples for each of these four groups: 1. Stretch, 2. Control for stretch, 3. Tachycardia, and 4. Control for tachycardia. Control samples and the respective intervention samples were matched and originated from the same patient. Due to a lower RNA yield than required for microarray hybridization it was first necessary to perform a single primer isothermal amplification (SPIA) using an Ovation Pico WTA System v2 kit (NuGen Technologies, Chesapeake Dr, Redwood City, CA, USA). A total amount of 20 ng RNA used for amplification which yielded 10 µg of SPIA cDNA in 20 µL volume. Microarray hybridization was conducted using GeneChip^™^ Human Gene 2.0 ST Arrays (Thermo Fisher Scientific, Wilmington, NC, USA). Data preprocessing and filtering was carried out with the Partek^®^ Genomics Suite^®^ software (v.6.6). Validation was performed by qPCR for selected genes. Bioinformatical assessment was done with Ingenuity^®^ Pathway Analysis software (Qiagen, Hilden, Germany).

### 4.4. Quantitative Real-Time PCR (qPCR)

Quantitative real-time PCR (qRT-PCR) was performed on a LightCycler^®^ 480 Instrument (Roche Applied Sciences, Penzberg, Germany). Validation of human microarray gene candidates from the same SPIA cDNA was performed using the TaqMan expression gene assay (Thermo Fischer Scientific) for the following genes: *CNN1* (Cat.#Hs00959434_m1), *SNRPN* (Cat.#Hs01374551_m1), *PLA2G7* (Cat.#Hs00965837_m1), *ADAM20* (Cat.#Hs01083178_s1), *MIR1183* (Cat.#Hs04273420_s1), and *GAPDH* (Cat.#Hs03929097_g1) along with TaqMan^™^ Fast Universal PCR Master Mix (2X), no AmpErase^™^ UNG (Thermo Fisher Scientific, Waltham, MA, USA). Verification of *MIR1183* gene expression in AFIB and DCM tissue was conducted after cDNA synthesis using the QuantiTect Reverse transcription kit (Qiagen, Hilden, Germany) with the TaqMan^®^ Fast Advanced Master Mix (Thermo Fisher Scientific, Waltham, MA, USA). GenEx 5.4.4. Software (MultiD, Gothenburg, Sweden) had been used to correct primer efficiency followed by 2^−ΔΔct^ relative quantification. The results are presented as fold change in gene expression normalized to control.

### 4.5. Statistical Analyses

Results are shown as bar charts representing mean ± SEM. Statistical significance between two groups with normal distribution, assessed via the Kolmogorov–Smirnov test, was determined by a paired-sample *t*-test. One-sample *t*-test was used when comparing qPCR fold-change to a control group with a pre-set value of 1. The microarray dataset was statistically evaluated by one-way ANOVA followed by Benjamini and Hochberg false discovery rate correction. *p*-values of <0.05 were considered as statistically significant.

## 5. Conclusions

Though stretch and tachycardia show distinct transcriptomic signatures in human atrial myocardium we identified a miR-1183 precursor gene with the highest up-regulation shared by both triggers. We verified its up-regulation status in acute and chronic remodeling, and end-stage heart failure and established a profile with consistent down-regulation of two putative downstream targets of miR-1183. Expression levels of miR-1183 might serve as a tissue biomarker for atrial remodeling and might have potential functional significance in cardiac disease. Future studies regarding its biomarker and functional role in cardiac disease will be of high interest and clinical relevance.

## Figures and Tables

**Figure 1 ijms-23-06962-f001:**
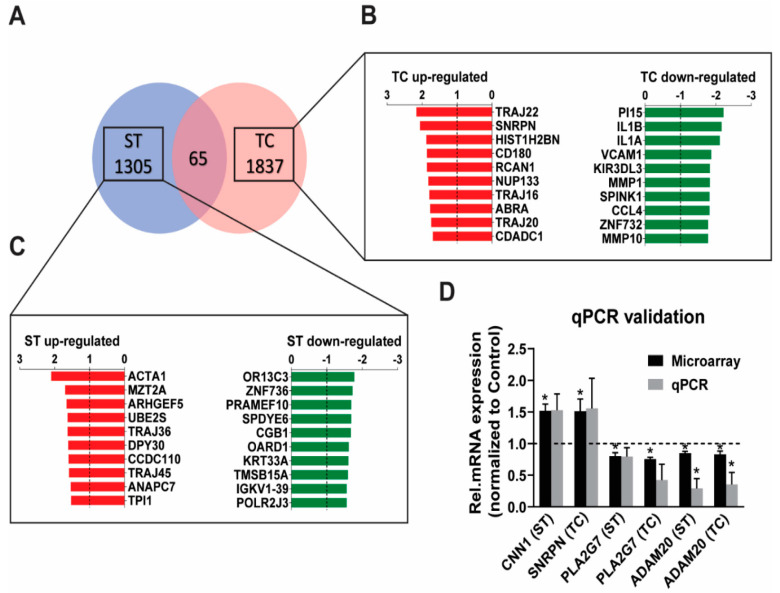
Genome wide expression characterization of stretch (ST) and tachycardia (TC) in human atrial trabeculae. (**A**) Microarray Venn diagram showing all genes regulated by ST or TC with *p* < 0.05 and fold-change cut-off of at least 1.2-fold up- or −1.2 down-regulated, *n* = 5 hearts per group. (**B**,**C**) A zoom-in of the top 10 genes (only protein coding ones) up-regulated (red) or down-regulated (green) after 6h of either stretch or tachycardia stimulation. (**D**) qPCR validation of some of the strongest regulated protein coding microarray genes with a determined function. Statistical difference determined by one-sample *t*-test. * *p* < 0.05, *n* = 4–5 hearts per group.

**Figure 2 ijms-23-06962-f002:**
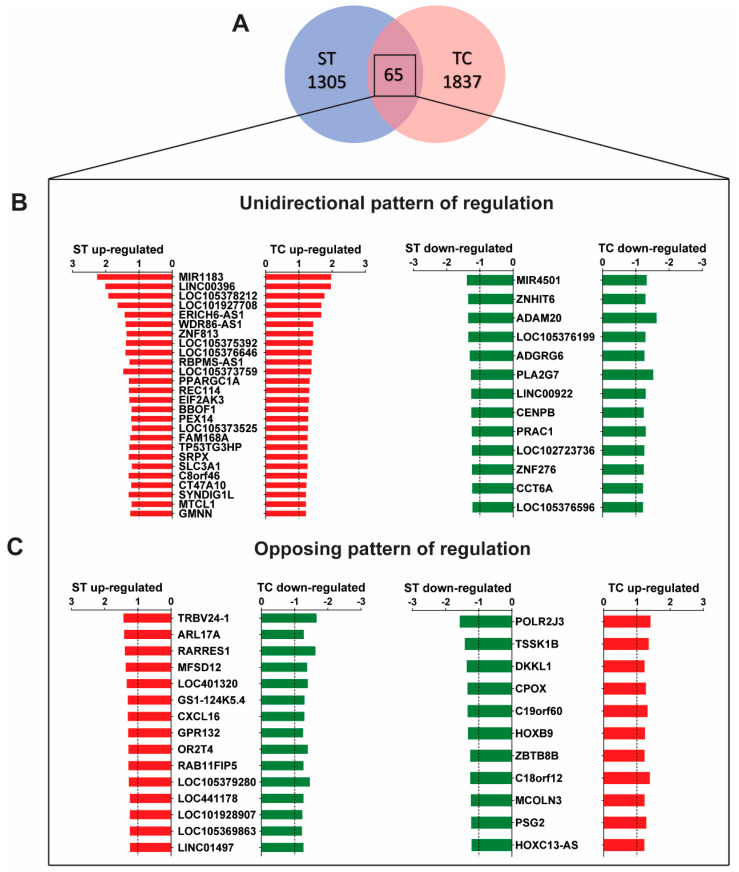
Commonly regulated genes by stretch (ST) and tachycardia (TC). (**A**) Microarray Venn diagram with cutoff values as specified in Figure 1. There are 65 regulated genes by both, ST and TC. (**B**) Out of this group there are 39 transcripts with the same direction and similar pattern of regulation. (**C**) 26 transcripts with opposing directed regulation. Statistical difference determined by one-sample *t*-test. *n* = 4–5 hearts per group.

**Figure 3 ijms-23-06962-f003:**
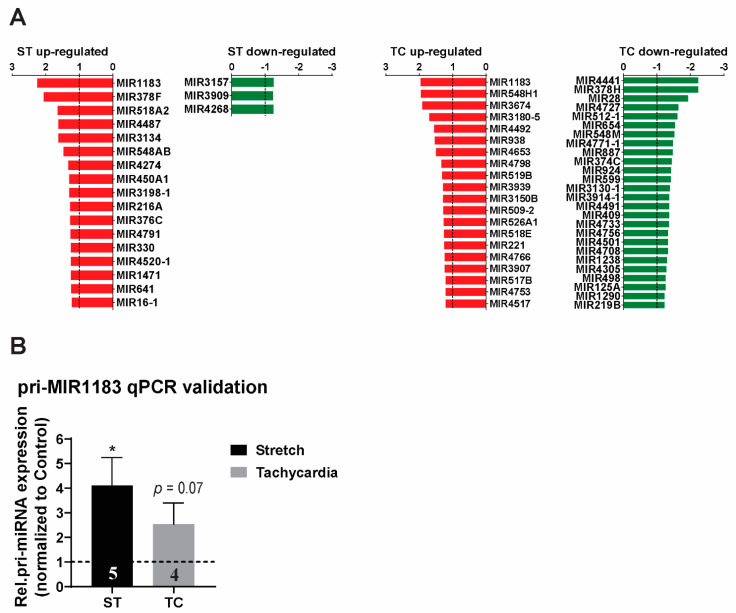
MicroRNA expression profile of ST and TC in human atrial trabeculae. (**A**) Microarray expression profile for precursors of miRNA in human atrial muscle strips regulated by ST or TC with *p* < 0.05 and fold-change cut-off of at least 1.2-fold up- or −1.2 down-regulated, *n* = 4–5 hearts per group. (**B**) pri-miR-1183 expression showed the highest fold change and was validated by qPCR in human atrial muscle strips. One-way ANOVA for microarrays and paired-sample *t*-test for qPCR, * *p* < 0.05, *n*-number of hearts per group is indicated within bar charts.

**Figure 4 ijms-23-06962-f004:**
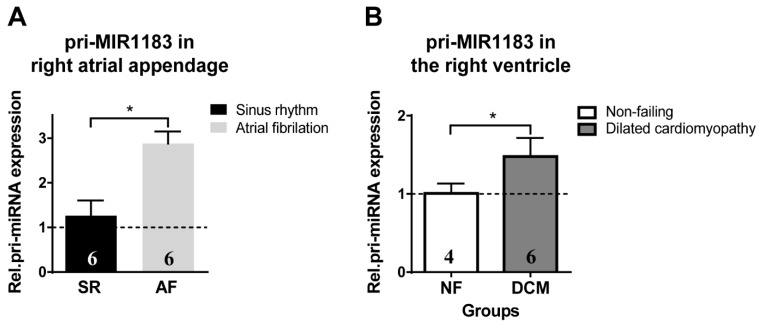
Chronic expression of pri-miR-1183 in healthy versus diseased atrial and ventricular human myocardium. (**A**) Increase in relative mRNA expression of pri-miR-1183 in right atrial appendage tissue from patients with normal sinus rhythm versus patients with persistent atrial fibrillation. (**B**) Increase in mRNA levels of pri-miR-1183 in right ventricular myocardial samples from the following heart groups: non-failing donor hearts (NF) and hearts with dilative cardiomyopathy (DCM). The dashed line represents the baseline expression. Statistical difference determined by one-sample *t*-test. * *p* < 0.05, *n*—number of hearts per group is indicated within bar charts.

**Figure 5 ijms-23-06962-f005:**
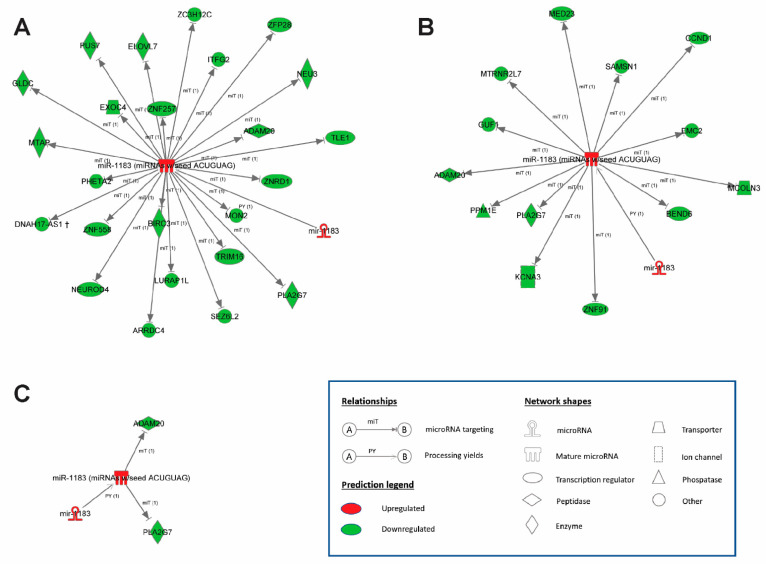
MicroRNA–mRNA network displaying the direct regulation of target genes by miR-1183. Up-regulation of miR-1183 and down-regulation of target genes following (**A**) tachycardia and (**B**) stretch. (**C**) Commonly regulated miR-1183 and target genes *ADAM20* and *PLA2G7*. Ingenuity^®^ Pathway Analysis, *n* = 4–5 hearts per group.

**Table 1 ijms-23-06962-t001:** Baseline characteristics of patients that underwent surgical interventions providing atrial muscle strip specimens.

Demographic Parameters/Medical History		Medication (%)	
Gender m/f (%)	56.6/44.4	ß-blockers	44.4
Age (y ± SD)	66.8 ± 9.4	ACE Inhibitors	44.4
BMI (% ± SD)	29 ± 6.4	AT1 Antagonist	22.2
SR (%)	100	MRA	0
AFIB	0	Statins	77.7
Paroxysmal AFIB	0	Ca^2+^ Antagonist	44.4
EF (% ± SD)	54.9 ± 9.4		
CABG (%)	77.8		
AVR (%)	11.1		
MVR (%)	11.1		

Abbreviations: BMI—Body mass index, SR—sinus rhythm, AFIB—atrial fibrillation, EF—ejection fraction, CABG—coronary artery bypass grafting, AVR—aortic valve replacement, MVR—mitral valve replacement, ACE—angiotensin-converting-enzyme, AT1—angiotensin 1, and MRA—mineralocorticoid receptor antagonist. *n* = 9.

**Table 2 ijms-23-06962-t002:** Baseline characteristics of sinus rhythm (SR) and atrial fibrillation (AFIB) groups of patients that underwent surgical interventions and provided atrial muscle strip specimens.

Groups	SR	AFIB
	Demographic parameters	
Gender m/f	1/5	4/2
Age (y ± SD)	66 ± 10	73.7 ± 9.8
BMI (% ± SD)	29.2 ± 3.5	24.8 ± 2.9
	Cardiac function	
EF (%± SD)	60.8 ± 9.2	56.2 ± 1.5
LVEDD	49.2 ± 2.4	57.2 ± 3.1
RVEDD	33.8 ± 1.3	33.3 ± 0.9
IVS	11.8 ± 0.9	12.8 ± 0.4
LA major axis	62.8 ± 3.6	70.4 ± 2.3
	Comorbidities (%)	
Hypertension	83	100
Diabetes	17	0
CABG	17	67
AVR	17	17
MVR	100	100
	Medication (%)	
ß-blockers	67	83
ACE Inhibitors	50	33
AT1 Antagonist	17	33
MRA	17	17
Statins	50	33
Ca^2+^ Antagonist	17	0

Main groups: SR—sinus rhythm, AFIB—atrial fibrillation, Parameters: BMI—body mass index, EF —ejection, LVEDD—left ventricular dilation diameter, RVEDD—right ventricular dilation diameter, IVS—interventricular septum, LA—left atria, CABG—coronary artery bypass grafting, AVR—aortic valve replacement, MVR—mitral valve replacement, ACE—angiotensin-converting-enzyme, AT1—angiotensin 1, and MRA—mineralocorticoid receptor antagonist. *n* = 12.

**Table 3 ijms-23-06962-t003:** Baseline characteristics of dilated cardiomyopathy (DCM) and non-failing (NF) ventricle donors.

Groups	NF	DCM
Gender m/f (%)	50/50	100/0
Age (y ± SD)	63 ± 16.3	59 ± 14
BMI (% ± SD)	25.4 ± 5.1	25.8 ± 2
SR (%)	75	33.3
AFIB (%)	25	66.7
EF (%± SD)	58 ± 0.13	22.5 ± 6.12
CHD	0	16.7
Hypertension	25	83.3
Diabetes	0	33.3
Hyperlipidemia	0	50
Paroxysmal AFIB	25	33.3
CABG	0	0
AVR	0	0
MVR	0	0
ß-blockers	25	33.3
ACE Inhibitors	25	50
AT1 Antagonist	0	0
MRA	0	50
Statins	0	16.7
Ca^2+^ Antagonist	0	0

Main groups: NF—non failing ventricles, DCM—dilated cardiomyopathy ventricles. Parameters: BMI—Body mass index, SR—sinus rhythm, AFIB—atrial fibrillation, EF—ejection fraction, CABG—coronary artery bypass grafting, AVR—aortic valve replacement, MVR—mitral valve replacement, CHD—coronary heart disease, ACE—angiotensin-converting-enzyme, AT1—angiotensin 1, and MRA—mineralocorticoid receptor antagonist. *n* = 24.

## Data Availability

Microarray data files are uploaded to the GEO repository under the series record GSE135166 and are publicly available. The data supporting findings of this study are available from the corresponding authors upon reasonable request.

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
