# Peer review of "miR-1183 Is a Key Marker of Remodeling upon Stretch and Tachycardia in Human Myocardium"

_ijms, 2022, doi:10.3390/ijms23136962_

Round 1
Reviewer 1 Report
Dr. Djalinac et al investigated the human heart samples and found that miR-1183 was increased during stretch and tachycardia in human atrial myocardium and could be a key marker of early remodelling. Although it is very interesting, it does not reach the level of being published in the International Journal of Molecular Science.
It seems to be a very valuable finding that miR-1183 can be a marker. However, it is ethically difficult to perform a cardiac biopsy in all cases in order to confirm these expressions in the atrial muscle. Please indicate the analysis results of the blood sample in the result section.
Author Response
Reviewer(s)' comments to author: Reviewer 1
- “Dr. Djalinac et al investigated the human heart samples and found that miR-1183 was increased during stretch and tachycardia in human atrial myocardium and could be a key marker of early remodelling. Although it is very interesting, it does not reach the level of being published in the International Journal of Molecular Science.”
Response: We thank the reviewer for this favorable but also critical comment and would like to elucidate the merit of our study: We performed an unbiased screening approach by microarray analysis in an acute experimental setup in isolated human myocardium which is demanding to perform and unique in literature. We found miR-1183 as highest upregulated transcript and provided robust evidence that this microRNA is regulated acutely by two independent triggers and has regulated downstream target genes in both. Furthermore, we confirmed the role of miR-1183 in chronic remodeling utilizing atrial and ventricular human myocardial tissue, which proved the translational efficacy of the study and identified miR-1183 as potential tissue biomarker for atrial remodeling with clinical implications. We are convinced that the scientific merit of the current study is high and hope that with this new light on our findings and the improvements of the revised version both reviewers will now find the manuscript suitable for publication in IJMS.
- “It seems to be a very valuable finding that miR-1183 can be a marker. However, it is ethically difficult to perform a cardiac biopsy in all cases in order to confirm theses expressions in the atrial muscle. Please indicate the analysis results of the blood sample in the result section.”
Response: We thank the reviewer for this comment, and we also thank the reviewer for raising the concern about the confirmation of our findings and the feasibility of using miR-1183 as a biomarker and the value of blood samples for this purpose. Regarding the confirmation we would like to point out that our data already provides confirmation of miR-1183 regulation in surgical tissue samples of chronic human cardiac disease of atrial and ventricular origin. Using samples of the right atrial appendage during cardiac surgery might indeed provide a future clinical implication for using miR-1183 as tissue biomarker, which is more obvious than biopsies. In those surgical samples miR-1183 expression levels might then be correlated with the risk of developing atrial fibrillation after cardiac surgery or other outcome parameters which needs to be shown in further studies.
But we fully agree with the reviewer´s further comment that confirming our tissue derived data by blood-based analyses is intriguing and warrants further studies. For the patients of our presented study matched blood samples collected in a standardized fashion were not available. We, therefore refrained from performing these analyses but provided an in-depth tissue characterization. However, we plan to perform measurements in blood samples in a prospectively designed follow up study.
Reviewer 2 Report
Djalinac and colleagues miR-1183 is a key marker of early remodeling upon stretch and tachycardia in human atrial myocardium. The manuscript is interesting and well written. The design of the study is straightforward.
Some minor problems should be corrected:
The title is misleading as they found upregulation also in atrial samples from patients with persistent tachycardia and in ventricular DCMP samples.
Abstract: “upregulate miR-1183 and its downstream targets” – As far as I can see the downstream targets are downregulated.
Line 50: stretch is the cellular correlate of pressure overload – is incorrect.
Chapter 2.1.: A comparable setup for human atrial trabeculae has been described many years ago and should be referenced (doi: 10.1007/BF00240044).
Line 90: How many experiments were performed in total?
Chapter 2.3.: Are the samples the same as in 2.2.?
Chapter 2.4.: How normal distribution was tested?
Line 138: “genes with unassigned function (reminder)” is unclear
Table 2: The abbreviations SR, AFIB should be included in the table headline. The same applies to Table 3.
Line 356: “strong correlation with the chip findings” please rephrase chip findings.
Line 374ff is completely speculative. Please remove.
Author Response
Reviewer(s)' comments to author: Reviewer 2
“Djalinac and colleagues miR-1183 is a key marker of early remodeling upon stretch and tachycardia in human atrial myocardium. The manuscript is interesting and well written. The design of the study is straightforward. Some minor problems should be corrected”
Response: We thank the reviewer for the favorable comments and the opportunity to address all specific points with our revised version.
- “The title is misleading as they found upregulation also in atrial samples from patients with persistent tachycardia and in ventricular DCMP samples”
Response: We agree that the previous title was focusing on only part of our results, we adapted the title accordingly in the revised version.
- “Abstract: “upregulate miR-1183 and its downstream targets” – As far as I can see the downstream targets are downregulated”
Response: We thank the reviewer for pointing out this unintentional inaccuracy. We edited the sentence in the revised version.
- “Line 50: stretch is the cellular correlate of pressure overload – is incorrect”
Response: We corrected the sentence in the revised version and would like to add that we used a form of stretch in our experiments that consisted of high preload but also increased afterload by isometric contraction. Thus, this form of stretch should not only correspond to volume overload but also to acute pressure overload, since in the latter PV curves in vivo are shifted to both, higher pressure and higher volume (doi:10.1007/s11340-020-00643-z). (line 51-53)
- “Chapter 2.1.: A comparable setup for human atrial trabeculae has been described many years ago and should be referenced (doi: 10.1007/BF00240044)”
Response: We are thankful for the reviewer’s suggestion. The reference was added to chapter 4.1., line 366 (previously 2.1., order changed according to the journal’s requirements).
- “Line 90: How many experiments were performed in total?”
Response: The Microarray experiments were performed on a total of 20 arrays (10 for stretch-related and 10 for tachycardia-related samples). Each of the subsets corresponding to a particular triggering insult had an equal number of arrays for control and intervention (5 for control and 5 for intervention). In the revised manuscript we described the procedure with more detail in chapter 4.3, line 383-386. (previously 2.3).
- “Chapter 2.3.: Are the samples the same as in 2.2.?”
Response: Yes, the samples were identical. All samples described in the dataset were subjected to the same RNA purification method as in new chapter 4.2. (previously 2.2). The samples used to verify the microarray findings by qPCR are the same as the once used to perform microarray hybridization, while miR-1183 expression was also assessed in other tissue samples that reflect chronic remodeling. This discrepancy was now made clear in chapter 4.2, lines 376;381 and 4.4, lines 399;405 (previously part of chapter 2) following the reviewer’s remark.
- “Chapter 2.4.: How normal distribution was tested?”
Response: Normal distribution was tested using the Kolmogorov-Smirnov test, and this information is now included in the respective chapter on statistical analysis, chapter 4.5, line 413.
- “Line 138: “genes with unassigned function (reminder)” is unclear”
Response: We thank the reviewer for pointing this out. The microarray transcripts are classified in four major groups as protein coding, non-coding and transcripts with a predicted function which is either protein coding or non-coding. Additionally, the array provides information on sequences that are mapped within the genome, however, have not an assigned function. We incorporated this information in a clearer manner in the revised version of the manuscript (Result part, now 2.1, line 84)
- “Table 2: The abbreviations SR, AFIB should be included in the table headline. The same applies to Table 3.”
Response: This will definitely strengthen the readability and clarity. We followed up this suggestion by implementing the abbreviations for the full phrase “Sinus Rhythm” (SR) and “Atrial Fibrillation” (AFIB) to the headline of Table 2, line 158 and similarly “Non-Failing” (NF) and “Dilated Cardiomyopathy” (DCM) to the headline of Table 3, line 168.
- “Line 356: “strong correlation with the chip findings” please rephrase chip findings.”
Response: We rephrased the sentence (line 302).
- “Line 374ff is completely speculative. Please remove.”
Response: We thank the reviewer for raising this concern. Indeed, the cardiac upstream regulators of miR1183 are unknown. In the initial manuscript we attempted to offer at least some hypotheses about potential upstream mechanisms to the reader, but we agree that these were entirely speculative in nature. Therefore, we removed this part of the paragraph (line 320).
Round 2
Reviewer 1 Report
The authors responded fully to our comments.